# Valorization of *Delonix regia* Pods for Bioethanol Production

Zafar Iqbal [1,]* , Adarsh Siddiqua [2], Zahid Anwar [2] and Muhammad Munir [3]

1   Central Laboratories, King Faisal University, Al-Ahsa P.O. Box 31982, Saudi Arabia
2   Department of Biochemistry and Biotechnology, University of Gujrat, Gujrat 50700, Pakistan
3   Date Palm Research Center of Excellence, King Faisal University, Al-Ahsa P.O. Box 31982, Saudi Arabia
*   Correspondence: zafar@kfu.edu.sa; Tel.: +966-580776536

**Abstract:** *Delonix regia* (common name: Flame tree) pods, an inexpensive lignocellulosic waste matrix, were successfully used to produce value-added bioethanol. Initially, the potentiality of *D. regia* pods as a lignocellulosic biomass was assessed by Fourier-transform infrared spectroscopy (FTIR), which revealed the presence of several functional groups belonging to cellulose, hemicellulose, and lignin, implying that *D. regia* pods could serve as an excellent lignocellulosic biomass. Response Surface Methodology (RSM) and Central Composite Design (CCD) were used to optimize pretreatment conditions of incubation time (10–70 min), $H_2SO_4$ concentration (0.5–3%), amount of substrate (0.02–0.22 g), and temperature (45–100 °C). Then, RSM-suggested 30 trials of pretreatment conditions experimented in the laboratory, and a trial using 0.16 g substrate, 3% $H_2SO_4$, 70 min incubation at 90 °C, yielded the highest amount of glucose (0.296 mg·mL$^{-1}$), and xylose (0.477 mg·mL$^{-1}$). Subsequently, the same trial conditions were chosen in the downstream process, and pretreated *D. regia* pods were subjected to enzymatic hydrolysis with 5 mL of indigenously produced cellulase enzyme (74 filter per unit [FPU]) at 50 °C for 72 h to augment the yield of fermentable sugars, yielding up to 55.57 mg·mL$^{-1}$ of glucose. Finally, the released sugars were fermented to ethanol by *Saccharomyces cerevisiae*, yielding a maximum of 7.771% ethanol after 72 h of incubation at 30 °C. Conclusively, this study entails the successful valorization of *D. regia* pods for bioethanol production.

**Keywords:** *Saccharomyces cerevisiae*; *Delonix regia*; biomass; pre-treatment; acid hydrolysis; fermentation; response surface methodology

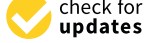



## 1. Introduction

The upsurge in global population and the development of human society have not only exacerbated food and energy demands but also resulted in environmental issues and global warming. In order to meet the everyday increasing food requirements of the growing population, agricultural practices are being intensified to produce more food, resulting in the production of agro-cellulosic biomass, which is becoming a burgeoning problem due to inefficient and inconsistent disposal and poor management practices [1]. The global biomass waste generation is expected to be around 140 gigatonnes per annum, with negative environmental repercussions. Most biomass wastes are left to decompose in the field or burnt, causing substantial environmental pollution [2]. The world population has become overly reliant on fossil fuels and their derivatives to meet energy demands. The extensive use of fossil fuels and their products results in the emission of greenhouse gases such as methane, carbon, and nitrogen oxides [3] and has severely shaken the global climate. Notably, all these cues, food sustainability, human activities/population, and environment, are so intermingled that the escalation of one exacerbates the severity of the other, and this cycle continues indefinitely and thereby affects food security indirectly. It is signposted that by the middle of this century, at least a 40% reduction in greenhouse gas emissions is required to maintain the average increase by 1.5 °C [4]. Additionally, fossil fuels are on the verge of depletion. This scenario led to the finding of eco-friendly, cost-effective, sustainable, and renewable sources of energy. Biofuels ranked top in these attributes,

strictly fulfilling the aforementioned criterion, and have emerged as an appealing choice to meet global fuel requirements [5]. The most significant advantages of biofuel production are renewability, production of fewer toxins, and emission of less carbon. Furthermore, the bioconversion of lignocellulosic biomass into bioethanol is a sustainable approach with unrivaled advantages for substituting gasoline [6].

First-generation biofuels are produced from edible feedstocks such as potato, wheat, sugarcane, rice, and barley. The direct competition with food crops renders them unfit for biofuel production [7]. In second-generation biofuels, woody biomass and forest residues were used to cope with the problems associated with first-generation biofuels, but there are still limitations that exist [8]. In third-generation biofuels, seaweed (macroalgae) was used as it does not require a large and arable area and has a high growth rate [9]. However, some limitations exist, such as the highly volatile nature of microalgae-derived biofuel [10]. Despite different constraints, biofuels derived from lignocellulosic materials (second generation) remained the widely produced biofuels and have a considerable potential to substitute non-renewable gasoline.

It is expected that global industrial demand for ethanol will surge to reach 135.5 billion liters/annum by 2050 [11]. To meet such high demands, second-generation bioethanol production from lignocellulose, a more sustainable and greener biomass, is a promising choice. *Saccharomyces cerevisiae* (*S. cerevisiae*) is primarily recognized yeast for fermentation due to its high tolerance limits and high ethanol yields [12–14]. It is commercially utilized in the industrial-scale production of bioethanol [15]. *S. cerevisiae* can ferment the different kinds of hexose sugars only but not the pentose sugars [16].

Lignocelluloses are the most abundant biomass on the earth's crust, but the commercial production of bioethanol from them is harrowing due to their unique compositional factor and physicochemical structure. Cellulose is the main structural polysaccharide in plant cell walls, accounting for 30% to 50% of the dry weight of lignocellulosic biomass. Hemicelluloses are the second most abundant polysaccharide, accounting for 15–30% of the dry mass of lignocellulosic plants. Lignin, which accounts for 15–30% of the dry mass of lignocellulosic biomass, is the third most important component [17]. The presence of lignin sheaths and their unique chemical composition impede the hydrolysis of long-chain and highly packed polysaccharides into fermentable sugars. In order to circumvent this problem, pre-treatment is a prerequisite to release fermentable sugars, but this enhances the overall cost of the process and poses a barrier to large-scale bioethanol production [18]. To deal with the cost problem, a plethora of pre-treatment techniques, such as ammonia explosion, acid treatment, alkali treatment, biological, enzymatic, and a combination of these, have been developed. Nonetheless, pre-treatment with dilute acid at high temperatures effectively promotes the hydrolysis of lignocellulose [19,20]. Dilute acid substantially converts hemicelluloses into simple sugars, and the hydrolysis of hemicelluloses improves the digestibility of the residual cellulosic contents in the biomass [21]. In most cases of biomass pretreatment, high temperatures favor the hydrolysis and digestibility of cellulose in the presence of an acid. Subsequent to acid hydrolysis, enzymatic hydrolysis can complete the hydrolysis of remaining cellulosic biomass into fermentable sugars. Additionally, the optimization of pre-treatment conditions through Response Surface Methodology (RSM) and Central Composite Design (CCD) was achieved. Four different variables, such as acid concentration, temperature, reaction time, and amount of substrate, were optimized. The RSM employs statistical analysis to generate model equations for optimizing and predicting specific condition behavior. As a result, this can eventually lead to the use of a small number of resources, making any process highly economical.

*Delonix regia* (*D. regia*), common name; Flame tree; local name Gulmohar, is a flowering plant belonging to the leguminous family *Fabaceae* (subfamily *Caesalpinioideae*). It is grown as an ornamental tree in tropical and sub-tropical regions of the world, and its pods are considered agricultural waste [22]. Its flowers are large, with four spreading scarlets of pale yellow-red petals up to 8 cm in length and a fifth upright petal that is slightly larger and spotted with yellow and white. The size of pods can be up to 60 cm in length and 5 cm

in width. Young pods are green and flaccid but turn dark brown and woody on maturation. The seeds are small, weighing an average of 0.4 g (6.2 grains) [23].

The present study aimed to yield the highest concentration of reducing sugars from *D. regia* pods. The hydrolytic potential of *D. regia* pods in terms of glucose and xylose yields was demonstrated in this study using acidic and enzymatic pretreatment. The interoperable RSM technique was used to optimize pretreatment conditions. This study entails the successful valorization of *D. regia* pods for bioethanol through *S. cerevisiae*.

## 2. Materials and Methods

### 2.1. Biomass Collection and Estimation of Moisture Contents

*D. regia* pods were collected from the vicinity of the University of Gujrat, Pakistan (32°38′26″ N; 74°10′01″ E). The collected samples were washed thoroughly with tap water, dried, ground to a fine powder of uniform particle size, and finally stored at room temperature in a plastic bag for further use.

The oven-dry method was used to calculate the moisture content of *D. regia* pods. A 1.23 g of *D. regia* pods were dried in an oven at 90 °C for 72 hours (h) until a further loss in weight was not observed. The weight loss is the moisture content of *D. regia* pods was calculated as;

$$\text{Moisture content (\%)} = \frac{\text{Initial weight} - \text{Dry weight}}{\text{Initial weight}} \times 100$$

### 2.2. Fourier-Transform Infrared Spectroscopy Assay

A Fourier-transform infrared spectrophotometer ([FTIR]; IR Affinity-1, Shimadzu, Kyoto, Japan) was used to determine the structural properties of *D. regia* pods. The powder sample of *D. regia* was mixed with spectroscopic grade KBr powder and pelleted into 1 mm size. The scan was taken from 4000–650 $cm^{-1}$ and had a spectral resolution of 4 $cm^{-1}$, as described earlier [24].

### 2.3. Optimization of Variables by RSM and Statistical Analysis

The RSM was employed to optimize the variables for the pretreatment of *D. regia* pods to yield the maximum reducing sugars using CCD. The selected variable attributes were temperature (45–100 °C), incubation time (10–70 min [min]), amount of substrate (0.02–0.22 g), and acid concentration ($H_2SO_4$, 0.5–3%). The response was validated by generating 3D response surface plots with two parameters set simultaneously and observing the interactions of variables on glucose and xylose yield as well as lignin degradation. The response of different variables was explained in the quadratic regression model [25]. The relationship between the optimum values of each attribute was then determined at the *p*-value of 0.05. The F-test and *t*-test were applied to determine the statistical significance. MINITAB 17 software was used to compute analysis of variance (ANOVA) and equation coefficients [26].

### 2.4. Acid Hydrolysis

The powdered *D. regia* pods (0.02–0.22 g) were initially acid hydrolyzed by soaking them in RSM-optimized conditions of dilute $H_2SO_4$ (0.5–3% *w/v*) at different temperatures (45–100 °C) in a static incubator for 10–70 min. A total of 30 independent trials were conducted. After acid hydrolysis, the samples were filtered, washed with distilled water, and dried in an incubator at 40 °C. The filtrate was used to assess the released reducing sugars (glucose and xylose) and soluble and insoluble lignin (Sections 2.5 and 2.6).

### 2.5. Estimation of Reducing Sugars

In order to estimate the glucose and xylose after acid hydrolysis, the filtrate samples were centrifuged at 4000 rpm for 20 min. The supernatant was discarded, and precipitates

were used. The dinitro salicylic acid (DNS) reagent was used to estimate the glucose concentration, and the phloroglucinol method was used to estimate the xylose concentration [27].

For glucose estimation, 100 µL of pretreated biomass filtrate was taken into a test tube, then 1000 µL of DNS reagent was added before boiling the mixture for a few min until the color changed from pale yellow to orange. Later, 5 mL of distilled water was added, and absorbance was measured at 540 nm. For xylose determination, 5 µL of the sample was taken from the filtrate and mixed with 5 mL of phloroglucinol reagent. After boiling the mixture for 5 min, 10 mL of distilled water was added to the flask and allowed to cool at room temperature. The absorbance was measured at 554 nm using a spectrophotometer (PG80, UK).

### 2.6. Determination of Lignin Contents

To determine the lignin contents, all the *D. regia* samples, pretreated with RSM optimized variables and non-treated (controls), were acid hydrolyzed. A pre-weighed amount (0.016 g) of all pretreated and non-treated *D. regia* samples were added to a flask containing 0.15 mL of 72% $H_2SO_4$ and incubated at 30 °C for 4 h. Then 4.2 mL of distilled water was added to the flask and autoclaved for 2 h. The residues were filtered and washed with distilled water for 10–15 min to neutralize the acid. Subsequently, the residues were oven dried at 105 °C until constant weight and used for insoluble lignin. The filtrate was then used to calculate the soluble lignin by measuring its absorbance at 205 nm. The total soluble lignin percentage was determined using the following formula [28,29].

$$\text{Total soluble lignin (\%)} = \frac{\text{Abs} + W_1}{W_2} \times 100$$

where:

    Abs = absorbance of soluble lignin at 205 nm.
    $W_1$ = weight of total soluble lignin (in grams)
    $W_2$ = weight of biomass (which was 0.016 g)

### 2.7. Enzymatic Hydrolysis

An indigenous cellulase enzyme produced by *Aspergillus tubingensis* via pre-optimized solid-state fermentation of corn stover [30] was used to augment the hydrolysis of lignocellulose of acid-hydrolyzed *D. regia* pods. For enzymatic hydrolysis, an enzyme activity of 74 filter per unit ($FPU \cdot mL^{-1}$) was determined [31] and added to a flask containing 2 g of the substrate in 100 mL of sodium acetate buffer (pH = 6). 74 $FPU \cdot mL^{-1}$ of the cellulase enzyme were added in different volumes; 0.5 mL ($0.5 \ U \cdot mL^{-1}$), 1 mL ($1 \ U \cdot mL^{-1}$), 1.5 mL ($1.5 \ U \cdot mL^{-1}$), 3 mL ($3 \ U \cdot mL^{-1}$), and 5 mL ($5 \ U \cdot mL^{-1}$). Flasks were incubated at 50 °C for different time periods. The samples were taken after 1.5 h, 3 h, 24 h, 48 h, and 72 h, and the glucose was determined through the DNS method [27]. In order to prevent bacterial contamination, ~75 mg of Augmentin was used in each flask. At the same time, the sample with the highest amount of glucose was used for the subsequent fermentation process.

### 2.8. Media and Inoculum Preparation

For inoculum preparation, *S. cerevisiae* (Rossmoor Food Products, Karachi, Pakistan) yeast was grown in Yeast Peptone Dextrose (YPD) growth media to carry out fermentation of the biomass. The YPD media were prepared by adding 2 g of casein peptone, 2 g of dextrose, and 1 g of yeast extract in 100 mL of distilled water [32]. Triplicate media samples were prepared in three independent flasks. All the flasks were sealed with cotton plugs and aluminum foil and then autoclaved at 121 °C for 30 min at 15 psi. The prepared inoculum was then used in the subsequent fermentation process.

### 2.9. Fermentation and Quantification of Ethanol

After enzymatic hydrolysis (Section 2.7), the filtrates of the pretreated samples carrying optimal glucose and pentose concentration were subjected to the fermentation process

to yield bioethanol. Then the filtrate was inoculated with 5 g of *S. cerevisiae* in different inoculum sizes (ranging from 1–5 mL), and fermentation was carried out in triplicates. Subsequently, the flasks of each treatment were incubated in a shaker at 30 °C, 120 rpm for 96 h. Fermented samples were taken under sterile conditions in a laminar airflow cabinet (Heraguard™ ECO Clean Bench, Thermo Fisher Scientific, Waltham, MA, USA) at 24, 36, 48, 72, and 96 h for the quantification of ethanol, as described earlier [25].

For the quantitative analysis of the produced bioethanol, the potassium dichromate method was used [33]. A 250 mL of dichromate solution (0.1 M of $Cr_2O_7^{-2}$ in 5M of $H_2SO_4$) was prepared by mixing 7.5 g of potassium chromate with 5 M $H_2SO_4$. Afterward, 3 mL of dichromate solution was taken in 250 mL beakers, and a falcon cap was placed in the center of beakers containing 300 μL of fermented solutions. The beakers were air sealed with parafilm and incubated for 30 min at room temperature, and the absorbance was measured at 590 nm, as described earlier [34].

## 3. Results and Discussion

### 3.1. Estimation of Moisture Contents

The moisture content in *D. regia* pods was determined as 9.48%, which was substantially higher than the MC% of previous studies, which reported 0.22% and 6.29% in *D. regia* pods [35,36]. The precise cause of the difference in MC% is far from a conclusion, but it can be attributed to varietal differences, habitat differences, or differences in axillary samples. The MC% of wheat straw was 4.2% [37], while other studies found 8.30% [38], 8.52% [39], and different lignocellulosic biomasses have 10–13% [40]. The moisture content provides a medium for nutrient transport, which is indispensable for the physiological and metabolic activities of microorganisms, resulting in a higher level of lignocellulosic material degradation [41,42]. Therefore, *D. regia* pods, with their higher moisture content, could serve as an unprecedented utility over other lignocellulosic matrices as excellent biomass for the production of next-generation biofuels.

### 3.2. Chemical Analysis of D. regia Pods by FTIR

FTIR is a widely used nondestructive analytical tool for the qualitative and quantitative identification of different types of chemical bonds (functional groups) present in chemical substances and lignocellulosic biomasses [43–46]. The FTIR absorbance spectra of *D. regia* pods revealed the presence of functional groups and vibration modes (Figure 1; Table 1). The absorption bands between 3200 and 3600 $cm^{-1}$ are attributed to the O-H stretching of alcohols, carboxylic acids, and hydroperoxides. The O-H stretchings of alcohols fall between 3650 and 3010 $cm^{-1}$. A peak at 3276 $cm^{-1}$ corresponded to vibration and free stretching of O-H groups in cellulose, hemicellulose, and lignin of *D. regia* pods [44,46–48]. The small peaks at 2915 $cm^{-1}$ demonstrated C-H asymmetric stretching, confirming the presence of cellulose in the biomass [44,47,49]. We found the $CH_2$ symmetric stretching in *D. regia* biomass at 2915 $cm^{-1}$, and similar stretching at 2914–2918 $cm^{-1}$ in cellulosic biomass has previously been demonstrated [44,49]. Other absorbance peaks in the 2000 and 2200 $cm^{-1}$ range were ascribed to C≡C in the alkynes in the *D. regia* pods. The peak at 1590 $cm^{-1}$ was accredited to skeletal vibration (C=C) in the phenolic ring of lignin [50]. The FTIR absorbance spectra of holocellulose and lignin found that the absorption positions at 1510 and 1600 $cm^{-1}$ are instigated by lignin. While absorption at 1730 $cm^{-1}$ by holocellulose [51], specifies the stretching of C=O in non-conjugated ketones, ester, and carbonyl groups.

The absorption spectrum at 1236 $cm^{-1}$ was attributed to C=O stretching of hemicelluloses and corroborated previous findings of C=O absorbance at 1244–1254 $cm^{-1}$ [44,49,52]. Likewise, C-O stretching at 1236 $cm^{-1}$ has been accredited to the guaiacyl unit of lignin [44]. Our results, however, contradicted a previous study that reported C-H asymmetric stretching at 2850 $cm^{-1}$ [53]. The absorbance peak at 1121 $cm^{-1}$ in *D. regia* pods had low molecular weight lignin fractions [47,52]. Similarly, at 1029 $cm^{-1}$, the C-H in-plane deformation for the guaiacyl unit and the C-O stretching of primary alcohols and cellulose was observed in *D. regia* pods. Other studies have found these stretching at 1028, 1032, and 1037 $cm^{-1}$,

which are in close proximity; the slight difference may be accredited to experimental conditions or the nature of the lignocellulosic biomass [46,47,53]. Nonetheless, another study has linked the peak at 1029 cm$^{-1}$ to the halogen (C-F) group [46]. Some sharp peaks were observed at 1935 and 1314, cm$^{-1}$, which were attributed to nitriles and carbonyl, respectively; this indicated the fortification of the *D. regia* pods by their combination. Similar findings have been demonstrated for the fortification of poplar biomass [44], corn cobs, and rice husks biomasses [46].

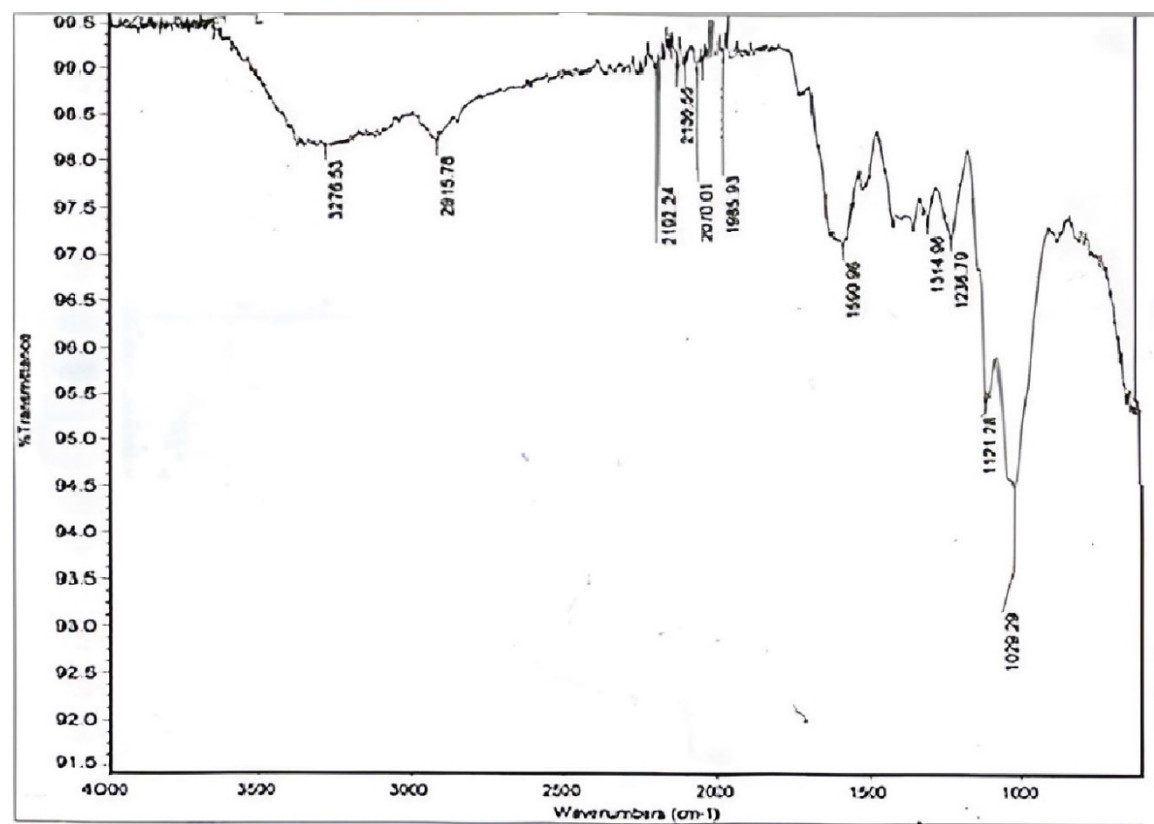

**Figure 1.** Fourier-transform infrared spectroscopy spectrum of *D. regia* pods.

**Table 1.** Fourier transform infrared (FTIR) band assignment of *D. regia* pods.

| Sr. No. | Wave Number (cm$^{-1}$) | Functional Groups | Bond |
|---|---|---|---|
| 1 | 3276.53 | Alcohol | O-H |
| 2 | 2915.78 | Alkane | C-H |
| 3 | 2192.24 | Alkynes | C≡C |
| 4 | 2135.55 | Alkynes | C≡C |
| 5 | 2070.01 | Alkynes | C≡C |
| 6 | 1935.93 | Nitriles | C≡N |
| 7 | 1590.96 | Alkenes | C=C |
| 8 | 1314.96 | Carbonyl | C=O |
| 9 | 1236.79 | Amines | N-H and C=O (Hemicellulose) |
| 10 | 1121.28 | Sulfoxide | S=O |
| 11 | 1029.29 | Halogen | C-F, C-H (guaiacyl unit of lignin) and C-O (primary alcohol and cellulose) |

### 3.3. Optimization of Physicochemical Parameters by RSM

The RSM (MINITAB 17) was employed to statistically optimize the interactive effects of four variables (two variables at a time) for the pre-treatment of *D. regia* pods. RSM is an efficient, cost-effective, and widely used method for designing and conducting experiments in biofuel production to achieve optimal conditions. RSM assesses the effects of different independent variables and their interactive effects with RSM-dependent variables, assisting in the reduction in experimental trials [54–58]. Additionally, it can predict results and generate 3D surface plots. RSM-based pre-treatment methods have been employed vastly to pretreat a wide range of lignocellulosic biomasses, including bamboo, corn stover, corn cobs, elephant grass, sugarcane, switchgrass, wheat straw, and *Vachellia nilotic* (Reviewed in [59]).

The minimum and maximum coded values (the lowest $-\alpha$, lower $-1$, mid 0, high $+1$, and the highest $+\alpha$) of four variables (amount of substrate, $H_2SO_4$ concentration, temperature, and time) were used. The RSM analysis created the Box-Behnken Design (BBD) matrix of the uncoded values of the variables with five levels of testing (Table 2). Design Expert v 7.0 software was used to analyze the experimental data [60].

**Table 2.** Coded and un-coded values of the variables for Box-Behnken Design.

| Sr. No | Variables | Un-Coded Values | | | | |
|:---:|:---:|:---:|:---:|:---:|:---:|:---:|
| 1 | Amount of substrate (g) | 0.02 | 0.04 | 0.1 | 0.16 | 0.22 |
| 2 | Acid (%) | 0.025 | 0.5 | 1.75 | 3 | 4.25 |
| 3 | Temperature (°C) | 45 | 60 | 75 | 90 | 100 |
| 4 | Time (minutes) | 10 | 20 | 45 | 70 | 90 |
| | | **Coded values** | | | | |
| | | $-\alpha$ | $-1$ | 0 | $+1$ | $+\alpha$ |

The rationality of the fitted model for glucose, lignin, and xylose production was analyzed through analysis of variance for the response surface models, and the F-test was opted to control the statistical significance (Tables S1–S9). Tables S1, S4 and S7 showed the analysis of variance for the response surface models of glucose, xylose, and lignin, respectively. The table shows that the regression models for glucose, xylose, and lignin yields were highly significant at confidence levels of 96.52%, 97.08%, and 96.52%, respectively, with very low probability values, i.e., (ca. 0.0001), and a high F-values of 29.72, 35.66, and 29.72, respectively. Model-fitted reliabilities of glucose, xylose, and lignin were calculated by determination coefficients ad were found to be 0.96 for all three. This indicates that around 96% of the variance is attributed to the variables. At the same time, the model could not elucidate only 4.0% of the overall variations. In addition, for the analyzed glucose, lignin, and xylose responses; $R^2$, adjusted and predicted, were all high, and the difference between both was less than 0.2, indicating that the model fit the data well (Tables S2, S5 and S8).

The response surface regression analysis for glucose production showed that two-way interaction analysis of all variables, except interaction between time and temperature, were non-significant at $p \leq 0.05$ (Table S1). For xylose production, the interaction of incubation time and the temperature was significant at $p \leq 0.05$ (Table S4). Likewise, for lignin production, the interaction of substrate and the temperature was non-significant at $p \leq 0.05$ (Table S7). This indicated the fitness and reliability of regression models. Our results corroborated an earlier report where the value of model fitness was 98.5% [25].

#### 3.3.1. Optimization of Interactive Variables for Glucose Production

For glucose production, 3D contour plots were inferred to yield the optimized interactive effects of the tested variables (Figure 2). An optimized relationship between temperature and incubation time revealed that longer incubation time and higher temperature resulted in higher glucose production. The glucose yield increased as hyperbolic

from lower (20 min) to higher (50 min) residence time and temperature (~80 °C). The interactive effect of both these variables could yield the highest glucose (~0.14 mg·mL$^{-1}$) at an incubation time of 50 min and 80 °C (Figure 2A).

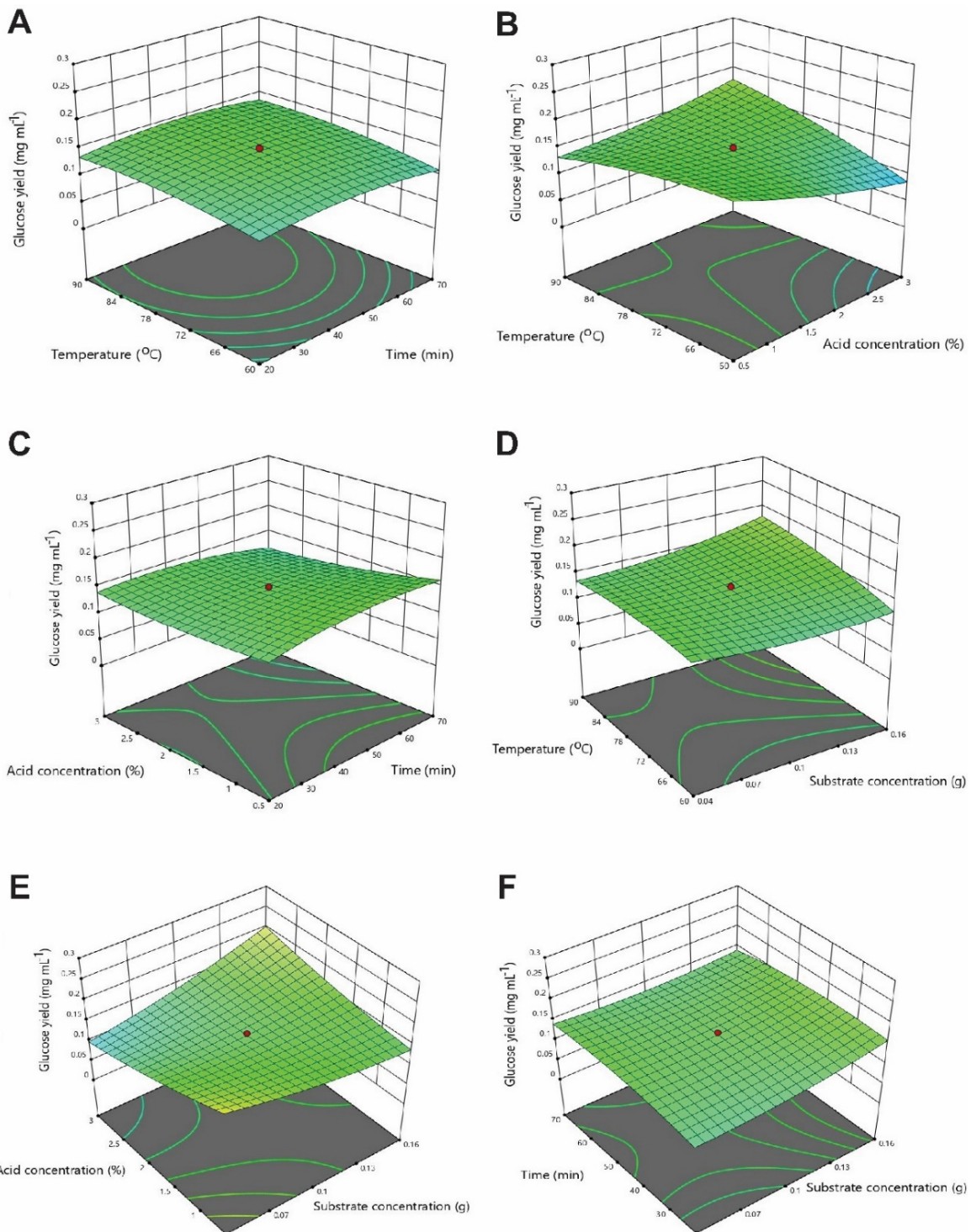

**Figure 2.** A 3D response surface plots for optimizing the interactive effects of temperature and acid concentration (**A**); temperature and time (**B**); acid concentration and time (**C**); temperature and substrate amount (**D**); acid concentration and substrate amount (**E**); and time and substrate amount (**F**) on glucose production.

The interactive effects of temperatures and $H_2SO_4$ revealed that a higher temperature (~90 °C) and a higher acid concentration (~3%) could yield the highest glucose (~0.17 mg·mL$^{-1}$) production (Figure 2B). Likewise, the interactive effect of temperature and substrate showed that the highest glucose (~0.17 mg·mL$^{-1}$) production could be yielded at a higher residence temperature (~90 °C) and substrate of 0.16 g (Figure 2D).

In Figure 2C, glucose yield increases as a hyperbolic plane with the increase in incubation time from 20–70 min. In contrast, acid concentration exhibited an opposite trend to incubation time. An interactive effect of acid concentration and incubation time revealed that the highest glucose production could be achieved by heating the biomass at a higher acid concentration (Figure 2C). In Figure 2E, both lower substrate- and acid concentrations could yield higher glucose. Nonetheless, the interactive effect showed the highest glucose (~0.2 mg·mL$^{-1}$) yield at 3% $H_2SO_4$ and an optimal amount of 0.16 g substrate (Figure 2E). The glucose yield increases linearly with an increase in substrate amount and incubation time; however, incubation of a high substrate amount (~0.13 g) at a high temperature (~45 °C) could lead to higher glucose (~0.15 mg·mL$^{-1}$) yield (Figure 2F).

### 3.3.2. Optimization of Interactive Variables for Xylose Production

A 3D contour plot for optimal xylose production (Figure 3) showed that incubation had no effect on the xylose yield, whereas lower (60 °C) and higher (90 °C) temperatures favored xylose production (Figure 3A). The interactive effect of temperature and time showed that 60 °C and 40–50 min of incubation could yield optimal xylose (~0.11 mg·mL$^{-1}$) production (Figure 3A).

The interactive effects of temperatures and acid concentrations revealed that a higher temperature and a lower acid concentration could yield higher xylose contents (Figure 3B). Likewise, the optimal xylose (~0.14 mg·mL$^{-1}$) could be released from the 0.04 of *D. regia* substrate at 90 °C (Figure 3D). The interactive effect of acid concentration and incubation time revealed that low acid concentration positively affected xylose production, whereas a very slight positive effect was observed on xylose production from 20–70 °C. The interactive effect of both these variables showed that the optimal xylose (~0.142 mg·mL$^{-1}$) could be yielded at 0.5% $H_2SO_4$ after 70 min of incubation time (Figure 3C). Figure 3E shows that xylose yield decreases slightly with increasing substrate amount and acid concentration, and 0.5% $H_2SO_4$ yields the highest xylose (~0.10 mg·mL$^{-1}$) from 0.04 g of substrate. In contrast, higher incubation time favored xylose production, and substrate amount had an almost negligible effect on xylose production. However, their interactive effect revealed that incubating 0.04 g of substrate for 70 min could yield the highest xylose (Figure 3F).

### 3.3.3. Optimization of Interactive Variables for Delignification

For percent soluble lignin estimation, 3D contour plots showed that the delignification of *D. regia* was affected severely at high temperatures, high acid concentration, and low residence time (Figure 4). In Figure 4A, the percent lignin decreases as incubation time increases, whereas time has a parabolic relation with delignification.

The higher acid concentration or temperature could lead to an increase in the percent lignin yield. The interactive effects of temperatures and acid concentrations showed that higher temperature (90 °C) and higher acid (3%) concentrations remarkably affect delignification (20%) (Figure 4B). Likewise, the optimal delignification could be yielded at a temperature of 60 °C and a substrate amount of 0.12 g; nonetheless, their interactive effect yielded the best results (~18%) at 90 °C and 0.016 g substrate (Figure 4D). Figure 4C reveals that delignification decreases linearly from 20–70 min and has parabolic relation from 0.5–3% $H_2SO_4$. The interactive effect of acid- and substrate-concentration showed the highest (~17%) lignin at 3% $H_2SO_4$ and 0.04 g of substrate, implying that low acid favors the delignification of *D. regia* pods (Figure 4E). The highest amount of lignin (~22%) could be yielded when 0.04 g or 0.16 g of the substrate was incubated for 20 min or 70 min, respectively (Figure 4F).

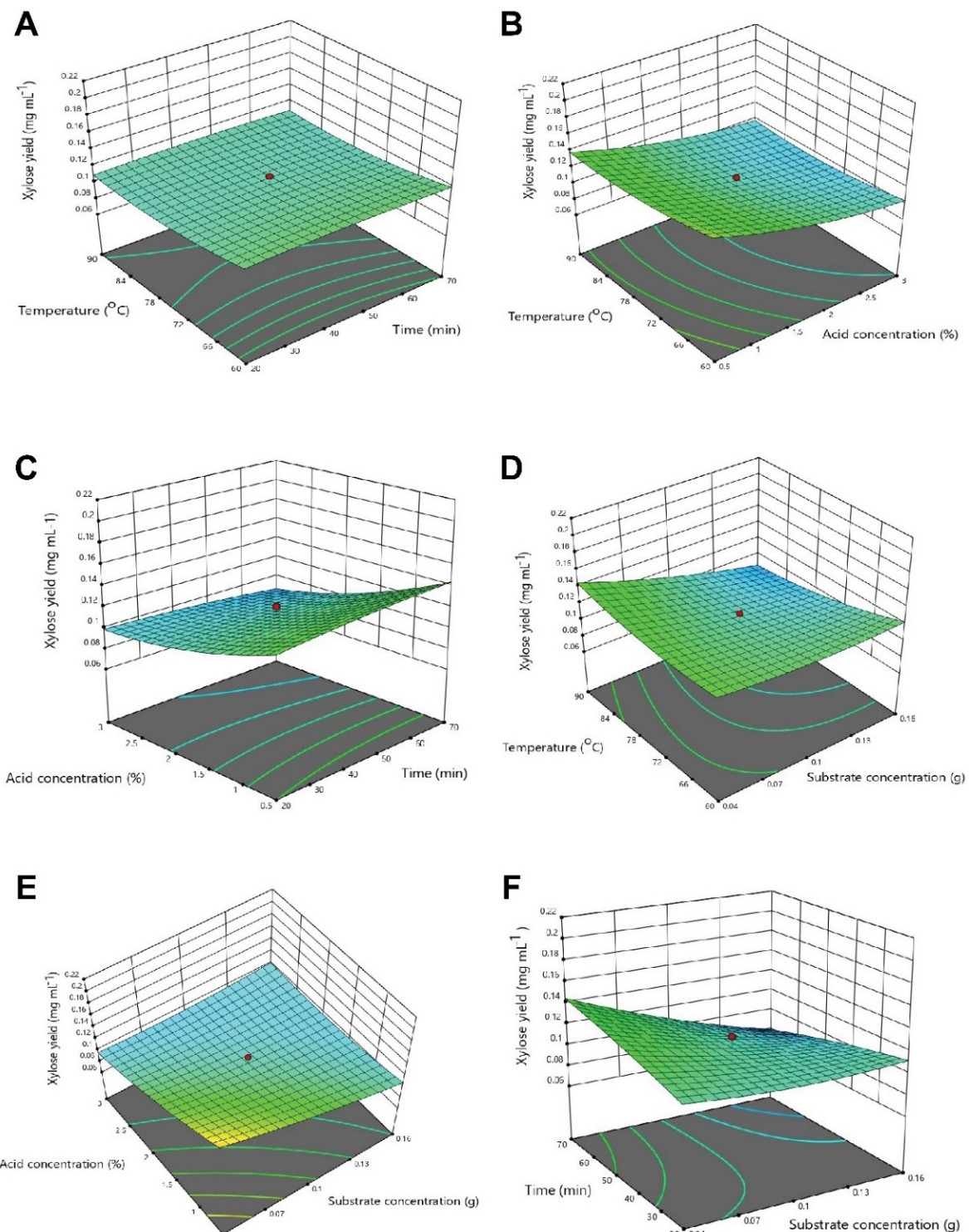

**Figure 3.** A 3D response surface plots for optimizing the interactive effects of temperature and acid concentration (**A**); temperature and time (**B**); acid concentration and time (**C**); temperature and substrate amount (**D**); acid concentration and substrate amount (**E**); and time and substrate amount (**F**) on xylose production.

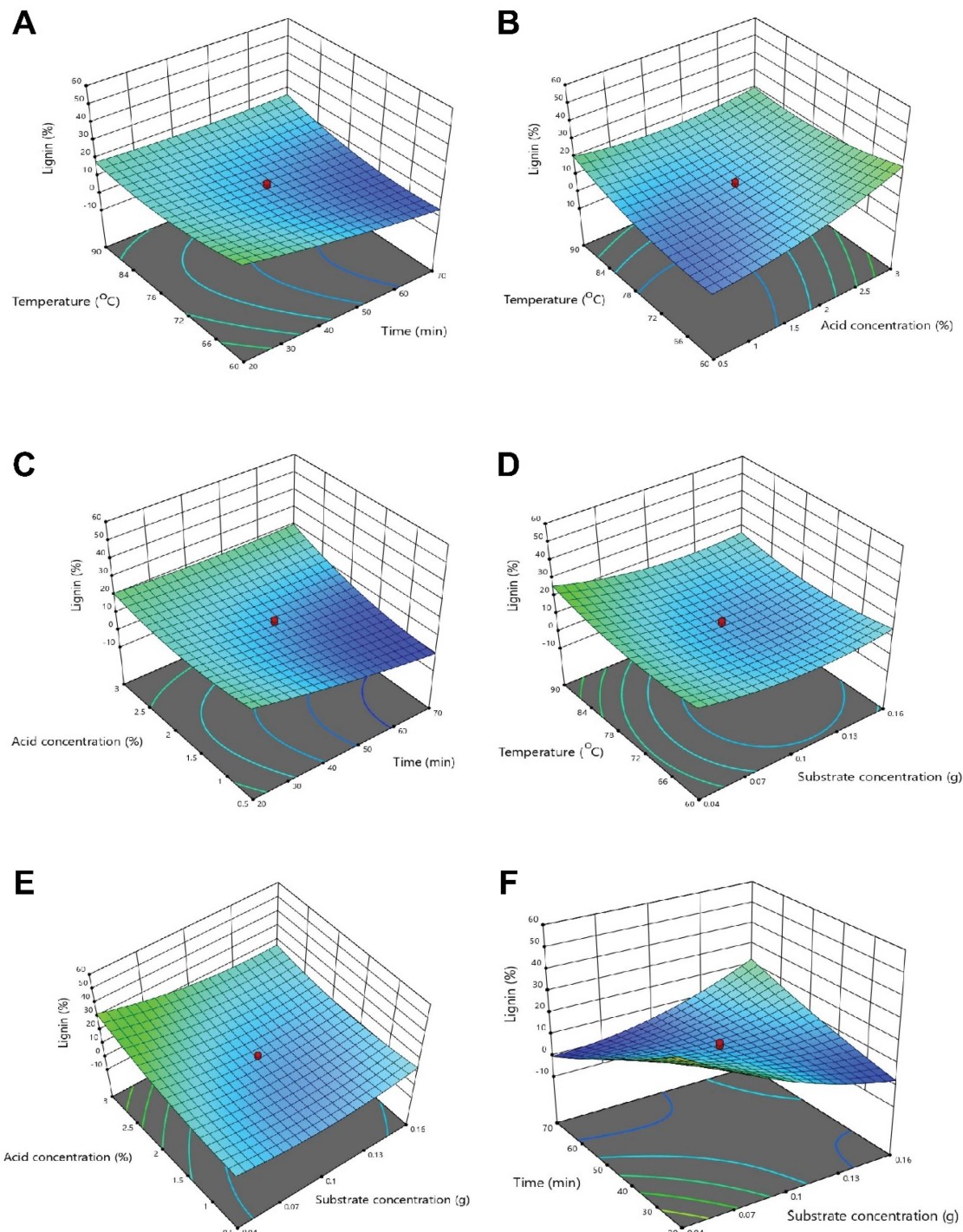

**Figure 4.** A 3D response surface plots for optimizing the interactive effects of temperature and acid concentration (**A**); temperature and time (**B**); acid concentration and time (**C**); temperature and substrate amount (**D**); acid concentration and substrate amount (**E**); and time and substrate amount (**F**) on delignification.

### 3.4. Experimental Testing and Optimal Experimental Values

The effect of four variables and their interaction on the production of reducing sugars (glucose and xylose) and delignification is presented (Table 3; Figure 5). RSM proposed 30 different experimental conditions (trials) of four variables and combinations of their different uncoded values to yield the optimal glucose, xylose, and lignin degradation (Table 3). The results of 30 trials revealed that trial 2 (with 3% $H_2SO_4$ at 90 °C for

70 min incubation time) produced the highest amount of glucose (0.296 mg·mL$^{-1}$) and xylose (0.477 mg·mL$^{-1}$) with the lowest amount of insoluble (residual) lignin (17.783%). Conversely, trial 27 produced the least amount of glucose (0.076 mg·mL$^{-1}$) and xylose (0.003 mg·mL$^{-1}$) with a higher amount of residual lignin (86.192%) (Table 3; Figure 5). These conditions of trial 2 were opted in the subsequent enzymatic hydrolysis, as it not only yielded a relatively low amount of insoluble lignin but also because of a higher amount of glucose and xylose. This implied that the interaction of four variables in trial 2 led to the swelling of the cellulose structure, thereby increasing the surface area and improving the digestibility of the residual lignocellulosic biomass [61].

**Table 3.** Effects of experimental variables (substrate amount, time, acid concentration, and temperature) on the yield of glucose, xylose, and insoluble lignin.

| Trial No. | Substrate (g) | Time (min) | Acid conc. (%) | Temperature (°C) | Glucose (mg·mL$^{-1}$) | Xylose (mg·mL$^{-1}$) | Insoluble Lignin (%) |
|---|---|---|---|---|---|---|---|
| 1 | 0.16 | 20 | 0.5 | 90 | 0.130 | 0.022 | 24.768 |
| 2 | 0.16 | 70 | 3.0 | 90 | 0.296 | 0.477 | 17.783 |
| 3 | 0.04 | 20 | 0.5 | 90 | 0.163 | 0.033 | 13.212 |
| 4 | 0.16 | 70 | 3.0 | 90 | 0.083 | 0.143 | 16.783 |
| 5 | 0.16 | 20 | 3.0 | 60 | 0.098 | 0.086 | 25.269 |
| 6 | 0.16 | 70 | 0.5 | 60 | 0.132 | 0.059 | 20.096 |
| 7 | 0.04 | 20 | 0.5 | 60 | 0.172 | 0.021 | 14.485 |
| 8 | 0.16 | 70 | 3.0 | 60 | 0.106 | 0.084 | 16.792 |
| 9 | 0.04 | 70 | 3.0 | 60 | 0.094 | 0.188 | 42.291 |
| 10 | 0.10 | 45 | 1.75 | 75 | 0.276 | 0.028 | 10.094 |
| 11 | 0.04 | 20 | 3.0 | 60 | 0.100 | 0.013 | 17.185 |
| 12 | 0.16 | 20 | 0.5 | 60 | 0.153 | 0.022 | 11.467 |
| 13 | 0.10 | 45 | 1.75 | 45 | 0.0935 | 0.011 | 49.292 |
| 14 | 0.22 | 45 | 1.75 | 75 | 0.0956 | 0.033 | 8.271 |
| 15 | 0.10 | 10 | 1.75 | 75 | 0.104 | 0.058 | 33.261 |
| 16 | 0.10 | 45 | 4.25 | 75 | 0.0935 | 0.039 | 34.231 |
| 17 | 0.10 | 45 | 1.75 | 75 | 0.108 | 0.023 | 18.184 |
| 18 | 0.02 | 45 | 1.75 | 75 | 0.134 | 0.1 | 26.291 |
| 19 | 0.10 | 95 | 1.75 | 75 | 0.102 | 0.024 | 3.91 |
| 20 | 0.16 | 20 | 3.0 | 90 | 0.102 | 0.02 | 38.849 |
| 21 | 0.16 | 70 | 0.5 | 90 | 0.077 | 0.12 | 6.292 |
| 22 | 0.16 | 20 | 3.0 | 90 | 0.119 | 0.242 | 37.292 |
| 23 | 0.10 | 45 | 1.75 | 100 | 0.127 | 0.099 | 6.246 |
| 24 | 0.10 | 45 | 0.25 | 75 | 0.119 | 0.016 | 7.332 |
| 25 | 0.10 | 70 | 3.0 | 90 | 0.098 | 0.033 | 5.651 |
| 26 | 0.16 | 70 | 0.5 | 60 | 0.129 | 0.009 | 51.192 |
| 27 | 0.10 | 45 | 1.75 | 100 | 0.076 | 0.003 | 86.192 |
| 28 | 0.10 | 70 | 1.75 | 75 | 0.102 | 0.036 | 37.242 |
| 29 | 0.04 | 70 | 0.5 | 60 | 0.212 | 0.022 | 49.209 |
| 30 | 0.16 | 70 | 0.5 | 60 | 0.083 | 0.001 | 8.207 |

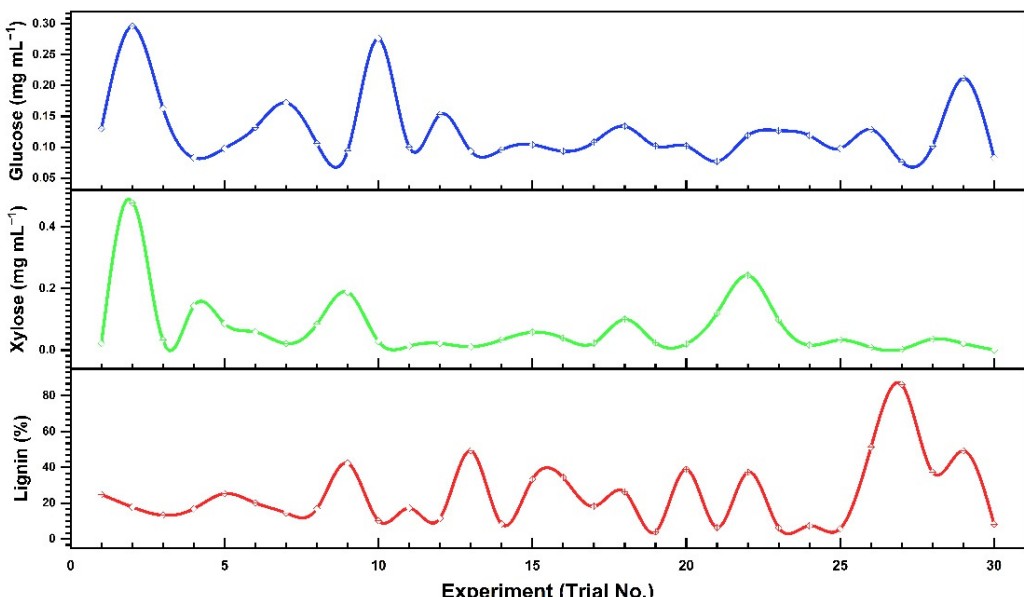

**Figure 5.** Glucose, xylose, and insoluble lignin were determined in 30 different trials.

Acid is a critical component in the pre-treatment process. Different types of inorganic and organic acids have been used for the pre-treatment of biomasses. Acid pretreatment is relatively inexpensive and has cascading effects on downstream processes. In this work, *D. regia* pods were pretreated with 3% $H_2SO_4$ resulted in the highest amount of reducing sugars. It may be due to the amorphous nature of the biomass, the dilute acid increased cellulose accessibility, and the release of fermentable sugars [62,63]. Predominantly, $H_2SO_4$ is regarded as a good choice for the pre-treatment process due to its greater ability to dissolve hemicellulose, alter the structure of lignin, and act as a catalyst. The dilute acid treatment had an unparalleled advantage over the concentrated acid and alkaline treatments because it is less toxic, less corrosive, and produces fewer byproducts of sugars [64–66]. Acid pretreatment methods change the lignin structure, increase hemicellulose solubilization, decrease cellulose crystallinity, and provide a larger surface area for the subsequent enzymatic hydrolysis step [65]. Furthermore, the acid itself hydrolyzes the biomass to fermentable sugars [67] and can release up to ~21.02% of reducing sugars from *Eulaliopsis binate* [68] and hydrolyze polysaccharides with minimum production of inhibitory compounds [69]. Dilute acid pre-treatment of *Bambusa spp.* with 5% acid for 30 min yielded 0.319 g·L$^{-1}$ of reducing sugar at 15% (*w/w*) [70,71], while pre-treatment of sugarcane biomass with 4.95% of acid at 80 °C for 375 min resulted in more than 99% saccharification and a concentration of 50.6 g·L$^{-1}$ monosaccharides [72]. Similarly, dilute acid hydrolysis of wheat straw for reducing sugar production at 106 °C, 0.98% $H_2SO_4$ for 45 min yielded 11.36 g·L$^{-1}$ of reducing sugars [58]. Pretreatment of bagasse pith with 4% $H_2SO_4$ for 90 min released the maximum glucose [73]. Pre-treatment of cobs, stalks, and maize leaves with dilute acid yielded 18.4 g·L$^{-1}$ (66.8%), 16.2 g·L$^{-1}$ (64.1%), and 11.0 g·L$^{-1}$ (49.5%) glucose yield, respectively [74]. Although 3% $H_2SO_4$ yielded the most glucose (0.296 mg·mL$^{-1}$) and xylose (0.2477 mg·mL$^{-1}$) from *D. regia* pods, these were the sub-optimal amount of reducing sugars to yield bioethanol. The precise reason is difficult to conclude, but we may speculate that structural feature of *D. regia* pods, such as crystallinity and surface area, were not sufficiently impacted during the acid pretreatment [75].

The current study found that a higher concentration of *D. regia* substrate (0.16 g) yielded the highest amount of reducing sugars. Several studies have achieved remarkable results in terms of yielding the highest amount of glucose, xylose, and maximum lignin degradation in a short time period through the acidic degradation of biomass at different temperatures. The substrate concentration of Oil palm fruit bunches revealed that higher substrate concentrations resulted in a higher glucose yield of 3.2 g·L$^{-1}$ [76]. When a high

concentration of aspen wood chips substrate (8%) was used, 85% of the cellulose could be hydrolyzed to glucose [77]. Substrate concentration behaves differently at different reaction times, and higher substrate concentration boosts glucose release as the reaction time increases [78].

Lignin provides rigidity to plant cell walls, protects them from physical and microbial breakdown, and renders the bio-polymeric structure to solubilization [79]. The strong bond between lignin and hemicellulose usually prevents easy access to the cellulose fraction during pre-treatment conditions [80,81]. The soluble % lignin contents (lignocellulosic biomass) were highest (~20%) in the *D. regia* pods at higher acid concentrations and temperatures, and this trend was consistent with previous studies [82,83]. Nonetheless, the results of 30 trials showed that polymeric lignin was not effectively degraded to monomeric sugars in many of them. In general, trials with higher reducing sugars contained less lignin, indicating that acid and temperatures boosted the conversion of polymeric lignocelluloses to low molecular weight phenolic compounds and monomeric carbohydrates [84]. Our results deviated from previous studies, which reported that delignification increased after biomass pre-treatment at a higher temperature (121 °C) and low acid concentration [85,86]. *D. regia* biomass was calcitrant at higher temperatures and acid concentrations, implying that *D. regia* pods may have a unique ratio of hemicellulose, cellulose, and lignin. This may also be accredited to the formation of pseudo-lignin due to carbohydrates dehydration and polymerization [87], or a prolonged incubation time adversely affects the delignification process by disrupting other bonds in lignin and producing other compounds [88]. Furthermore, longer incubation time could lead to dehydration of xylose, hampering the maximum possible xylose yield [89,90].

*3.5. Enzymatic Hydrolysis and Bioethanol Production*

The recalcitrant nature of celluloses and hemicelluloses rendered their complete breakdown. So, to release the maximum fermentable sugars, a second treatment could opt. Enzymatic saccharification has been proven to be a highly beneficial process for releasing the highest amount of glucose from lignocellulosic biomass [91]. The effective pre-treatment strategies overcome biomass's recalcitrant nature and provide an amenable substrate for subsequent enzymatic hydrolysis. To circumvent the recalcitrant nature of *D. regia* pods, a severe pretreatment with aggressive chemistry, followed by enzymatic hydrolysis, can be used [92]. In this study, the pretreated *D. regia* pods with the highest amount of glucose and xylose and comparatively less insoluble lignin were subjected to enzymatic hydrolysis with an indigenously produced cellulase enzyme. The results demonstrated that the enzymatic load of 5 U mL$^{-1}$ yielded the highest glucose concentration (55.57 mg·mL$^{-1}$) after 72 h of incubations (Figure 6A).

Notably, acidic pretreatment of *D. regia* pods merely produced 0.296 mg·mL$^{-1}$ of glucose, pinpointing the calcitrant nature of *D. regia* pods. Similar findings have earlier shown that two different types of pretreatment methods could yield optimal biodiesel production from *D. regia* pods [93]. Our results are in line with an earlier study, which reported that 50% of glucose was recovered from the rice hull after 48 h of enzymatic treatment [94]. Likewise, 80% of total glucose yield was achieved after pretreating wheat bran with 0.5–4% (*w/w*) of H$_2$SO$_4$ and then with 5% enzymatic load for 72 h [95]. Likewise, wheat bran pretreated with acid and subsequently with different enzymes, such as cellulase, xylanase, hemicellulase, and glucosidase, produced a higher concentration (95%) of total fermentable sugars [96].

After optimization of all physiochemical parameters, acidic pretreatment, and enzymatic saccharification of *D. regia* pods, the fermentation was carried out by *S. cerevisiae*. A gradual and logarithmic increase in ethanol concentration was observed from 24 h to 72 h of incubation, and then a slight decline was observed up to 96 h of incubation (Figure 6B; Table S10). The highest concentration of ethanol, 7.771%, was obtained after 72 h of incubation (Figure 6B). After 72 h of incubation, the yeast consumed the maximum amount of carbon source and approached the plateau phase, or the product entered into the

product inhibition phase. In this research, different incubation temperatures were used to yield the highest bioethanol concentration. The temperature has a noticeable influence on bioethanol production. So, it is indispensable to optimize this parameter [97]. Nimbkar and his colleagues fermented unsterilized sweet sorghum juice at 25, 30, and 35 °C, yielding the maximum ethanol (12.45%) at the 30 °C incubation temperature [98]. Our results are in harmony with Chongkhong et al. [66], who proclaimed high ethanol yield with increasing pH (4.4–5.9) and temperature of (27 to 36 °C), but the gradual decrease in yield was also reported with further increase in pH and temperature. Likewise, a high ethanol concentration was produced from the bagasse hydrolysate through *Pichia stipitis* BCC15191 at 30 °C, pH 5.5, after 72 h of incubation [99]. Our results deviated from Markou et al., who reported a fermentation yield is 56% by using *Antrosphira platensis* [100], and Ho et al., who reported a fermentation yield of 90% by using *Chlorella vulgaris* [101]. The reason behind the low production of bioethanol may be accredited to different microorganisms and their growth conditions.

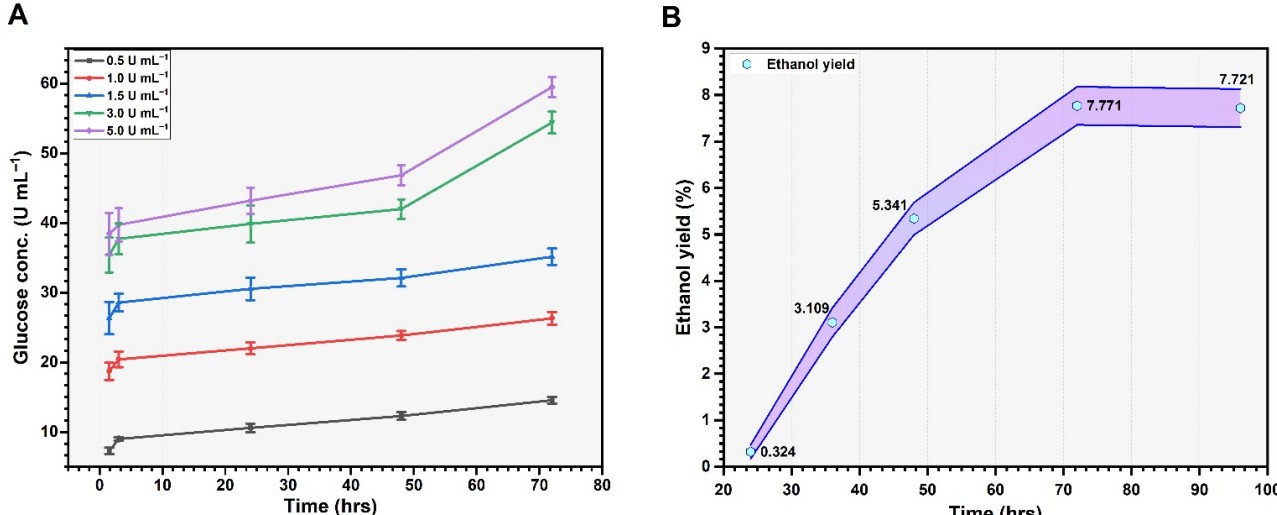

**Figure 6.** Glucose concentration (mg·mL$^{-1}$) yielded after the different time periods of enzymatic hydrolysis (**A**); and ethanol production (%) at different time-period (h) of fermentation (**B**). The graphs used the mean values of three replicates.

Although we successfully entailed the valorization of *D. regia* pods to bioethanol, however, to make this process more viable and economically acceptable, the cost of bioethanol production must not exceed the current gasoline price. This is achievable by improving the efficiency of *D. regia* pods processing technologies and could be the interest of futuristic study.

## 4. Conclusions

Conclusively, the FTIR analysis revealed that *D. regia* pods contained all the essential components that make it an excellent source of lignocellulosic biomass for bioethanol production. RSM and CCD are found to be effective tools for optimizing the physicochemical parameters for the pre-treatment of *D. regia* biomass. The interactions of different variables significantly impacted the hydrolytic potential of *D. regia* pods, as demonstrated by glucose and xylose yields with 3% H$_2$SO$_4$ at 90 °C for 70 min. However, acidic pretreatment of *D. regia* pods alone is insufficient to yield an optimal amount of fermentable sugars, so enzymatic hydrolysis was carried out to enhance the yield. In the fermentation, the highest amount of yielded ethanol was 7.771% using a commercial strain of *S. cerevisiae*. The current findings have opened the myriads of opportunities for utilizing *D. regia* pods for bioethanol production, which can be opted for on a pilot or industrial scale after making it more economical and cost-effective.

**Supplementary Materials:** The following supporting information can be downloaded at: https://www.mdpi.com/article/10.3390/fermentation9030289/s1, Table S1: Response surface regression analysis for glucose production and the interactive effects of substrate concentration versus substrate concentration (g), time (min), Acid concentration (%), and temperature (°C); Table S2: Model fit summary for Glucose production; Table S3: Final equation of glucose production in terms of actual factors; Table S4: Response surface regression analysis for xylose production and the interactive effects of substrate concentration versus substrate concentration (g), time (min), Acid concentration (%), temperature (°C); Table S5: Model fit summary for Xylose production; Table S6: Final equation of xylose production in terms of actual factors; Table S7: Response surface regression analysis for lignin degradation and the interactive effects of substrate concentration versus substrate concentration (g), time (min), Acid concentration (%), temperature (°C); Table S8: Model fit summary for lignin degradation; Table S9: Final equation of Lignin degradation in terms of actual factors; Table S10: Glucose recovery after enzymatic hydrolysis at different time-periods.

**Author Contributions:** Conceptualization, Z.A. and Z.I.; Investigation, A.S.; methodology, A.S., Z.A. and Z.I.; writing—original draft preparation, A.S., Z.I. and Z.A.; writing—review and editing, Z.I. and M.M. All authors have read and agreed to the published version of the manuscript.

**Funding:** The authors extend their appreciation to the Deputyship for Research and Innovation, Ministry of Education in Saudi Arabia, for funding this research work (Project# INST012).

**Institutional Review Board Statement:** Not applicable.

**Informed Consent Statement:** Not applicable.

**Data Availability Statement:** All the data related to this study is mentioned in the manuscript.

**Acknowledgments:** Authors are thankful to M. Adnan (University of Florida, USA) for critical reading and improving the language of the script.

**Conflicts of Interest:** There is no conflict of interest among the authors.

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
