# Peer review of "Valorization of Delonix regia Pods for Bioethanol Production"

_fermentation, doi:10.3390/fermentation9030289_

Round 1

Reviewer 1 Report

This manuscript studied the processing of Delonix Regia pods to bioethanol via acid-pretreatment, enzymatic hydrolysis and yeast fermentation. However, the manuscript has to be thoroughly revised before it can be published in Fermentation. Below are some suggestions for consideration:

The analysis and experiment in this study are not well related to the objective. For example, FTIR analysis is used for identifying surface functional groups on solid materials. What would be the purpose of FTIR analysis for the materials that would later be hydrolyzed and decomposed for sugar and ethanol production? Instead, a compositional analysis which could provide the chemical composition (glucan, xylan, etc.) of the materials should be done which, however, was not performed in this study.

Moisture content was discussed as important factor for the potential yield of ethanol with the material, which doesn’t make much sense. No matter in the lab study or in industry, the biomass is used based on dry matter basis. This means the initial solid material should be dried to get accurate estimation of loading rate in dry matter basis for the following process. Also, based on the equation below, bioethanol production is based only on the glucose content C6H12O6------2C2H5OH(l) +2CO2(g), which depends on the efficiency of hydrolysis.

The authors used RSM for optimizing pre-treatment conditions of the biomass. The glucose and xylose yield in the acid-pretreatment was mostly lower than 0.2 g/L while the enzymatic hydrolysis could yield glucose up to 60 g/L, what would be the purpose and value of optimizing acid-pretreatment for glucose and xylose?

It seems that the manuscript mainly focused on optimization on acid-pretreatment with only one page on enzymatic hydrolysis and ethanol fermentation. Therefore, the authors should discuss more on the economic value of optimization of acid-pretreatment, and the significance of reducing pretreatment time, acid usage, and increasing biomass loading, which could potentially reduce overall cost of lignocellulosic ethanol production. The whole process from lignocellulosic biomass pretreatment, enzymatic hydrolysis, and yeast fermentation have long been very commonly studied topics in the field of biofuel. However, the high-cost utilization of energy, chemicals, labors, equipment and the lower price of product (e.g. ethanol) make the whole process economically infeasible. Any breakthroughs from reducing the use of energy, chemicals, improvement of conversion efficiency, producing end-products of higher industrial value (e.g. higher alcohol, longer-chain fatty acids, polymers) could promote the lignocellulosic-bioproducts economical feasible.

Author Response

Submission of the revised script in the “Fermentation-MDPI”

Dear Reviewer,

I would like to submit our revised script, “Valorization of Delonix regia Pods for Bioethanol Production” to Fermentation.

I thank you, on behalf of all the co-authors, for helpful and detailed comments. We responded to every query and made tracked changes to the script. Please see below our comments and find the revised script attached herewith. 

Sincerely,

Zafar Iqbal, PhD

Response to Reviewers

The authors thank the reviewers for their helpful and detailed comments. Authors respond to each comment (in the red font, pls see below) and have made tracked changes to the script.

Reviewer 1

Comment 1. The analysis and experiment in this study are not well related to the objective. For example, FTIR analysis is used for identifying surface functional groups on solid materials. What would be the purpose of FTIR analysis for the materials that would later be hydrolyzed and decomposed for sugar and ethanol production? Instead, a compositional analysis which could provide the chemical composition (glucan, xylan, etc.) of the materials should be done which, however, was not performed in this study.

With regard to this FTIR analysis, the apparent purpose was to assess whether D. regia pods have enough hydrolysable materials to be employed in the study. So, it is a little astonishing that reviewer is asking “What would be the purpose of FTIR analysis for the materials that would later be hydrolyzed and decomposed for sugar and ethanol production?” Authors would like to say that its all about hydrolyzing lignocellulosic contents to fermentable sugars. In addition, the similar FTIR analysis has been performed for a number of other biomasses (Please see references 42-49). Thus, authors would prefer to state that this occurred to inform readers that this scenario is possible. We have thus modified the text but maintained the general statement of caution. Please see the revised version.

Comment 2. Moisture content was discussed as important factor for the potential yield of ethanol with the material, which doesn’t make much sense. No matter in the lab study or in industry, the biomass is used based on dry matter basis. This means the initial solid material should be dried to get accurate estimation of loading rate in dry matter basis for the following process. Also, based on the equation below, bioethanol production is based only on the glucose content C6H12O6------2C2H5OH(l) +2CO2(g), which depends on the efficiency of hydrolysis.

This is a good point and authors are agreed to the kind reviewer. Thus, the related statement has been modified. Please see the revised script.

Comment 3. The authors used RSM for optimizing pre-treatment conditions of the biomass. The glucose and xylose yield in the acid-pretreatment was mostly lower than 0.2 g/L while the enzymatic hydrolysis could yield glucose up to 60 g/L, what would be the purpose and value of optimizing acid-pretreatment for glucose and xylose?

We thank the reviewer for noting this and in general agree with this viewpoint. The precise reason to go for acidic pretreatment was its economy and widespread use to pretreat the lignocellulosic biomass. Nonetheless, no information was available to yield optimal amount of glucose and xylose from the pretreatment of D. regia pods. Hence, we opted for the most economical and widely used pretreatment method. Once we discovered that pretreatment of D. regia pods alone are insufficient to yield a sufficient amount of fermentable sugars then we subsequently went for enzymatic hydrolysis.

However, we found this point a good hit, so we added a similar statement in the results and conclusion.  

Comment 4. It seems that the manuscript mainly focused on optimization on acid-pretreatment with only one page on enzymatic hydrolysis and ethanol fermentation. Therefore, the authors should discuss more on the economic value of optimization of acid-pretreatment, and the significance of reducing pretreatment time, acid usage, and increasing biomass loading, which could potentially reduce overall cost of lignocellulosic ethanol production. The whole process from lignocellulosic biomass pretreatment, enzymatic hydrolysis, and yeast fermentation have long been very commonly studied topics in the field of biofuel. However, the high-cost utilization of energy, chemicals, labors, equipment and the lower price of product (e.g. ethanol) make the whole process economically infeasible. Any breakthroughs from reducing the use of energy, chemicals, improvement of conversion efficiency, producing end-products of higher industrial value (e.g. higher alcohol, longer-chain fatty acids, polymers) could promote the lignocellulosic-bioproducts economical feasible.

To us, this comment comprised of two parts, hence, we tried to respond it accordingly.

First, the reviewer pointed to add and discuss more about the economic value of optimization of acid-pretreatment, and the significance of reducing pretreatment time, acid usage, and increasing biomass loading, which could potentially reduce overall cost of lignocellulosic ethanol production. We have added more discussion regarding these points. Please see the revised script.

Second, about making the lignocellulosic-bioproducts economical and feasible. We entirely agreed with the reviewer; however, in this study our primary objective was to explore the potentiality of D. regia pods for bioethanol production. Making biofuel production more economical and feasible at pilot and industrial scale could be an interest of futuristic study.

Reviewer 2 Report

1. Please rewrite the abstract to better reflect the content of the work, in particular I find it important to clearly state that experiments are donein this work andused forthe analysis.

2. Make sure all abreviations are properly explained the first time they are mentioner.

3. More detiled description of your method and state-of-art is neede to better understand your work, methodology and findings.

4. Results need to be re-written. Add the proper discusion with reasoning 

5. In the experimental section, there is lack of real data, FILL IT UP.

6. English Language needs extensive chacking

Author Response

Submission of the revised script in the “Fermentation-MDPI”

Dear Reviewer,

I would like to submit our revised script, “Valorization of Delonix regia Pods for Bioethanol Production” to Fermentation.

I thank you, on behalf of all the co-authors, for helpful and detailed comments. We responded to every query and made tracked changes to the script. Please see below our comments and find the revised script attached herewith. 

Sincerely,

Zafar Iqbal, PhD

Response to Reviewers

The authors thank the reviewers for their helpful and detailed comments. Authors respond to each comment (in the red font, pls see below) and have made tracked changes to the script.

Reviewer 2

Comment 1.     Please rewrite the abstract to better reflect the content of the work, in particular I find it important to clearly state that experiments are donein this work andused forthe analysis.

The abstract has been revised thoroughly as per suggestion. Please see the revised version of the script.

Comment 2.     Make sure all abreviations are properly explained the first time they are mentioner.

Authors thoroughly checked the abbreviations, and found some oversights that have been corrected. Please see the revised version.

Comment 3.     More detiled description of your method and state-of-art is neede to better understand your work, methodology and findings.

Authors agree to the reviewer in part. We would rather state that the methodology section is generally self-explanatory, including relevant references for more detailed information, and it occurred to inform readers in general. Nonetheless, we tried to add more stuff, as per suggestion, to make it more understandable.

Comment 4.     Results need to be re-written. Add the proper discusion with reasoning

The result and discussion section has been revised. But authors would like to diligently request the reviewer to point out the missing information. Nonetheless, we found a few flaws in the discussion that have been corrected, and reasoning to some important points has been added as well.  

Comment 5.     In the experimental section, there is lack of real data, FILL IT UP.

All the presented data pertaining to coded and un-coded values of the variables in the RSM analysis was based on our REAL findings. Therefore, we opted all those to predict the un-coded values. Besides, authors did their best to FILL UP the data and revised the section. Hopefully, this will suffice the query.

Comment 6.     English Language needs extensive chacking

A native English speaker has improved the script's language and he has been acknowledged for his contribution.

Round 2

Reviewer 1 Report

The language of the manuscript has been improved. However, the commonts were simply addressed by saying "please see the revised version" without giving a direct location in the manuscript where it has been modified. Below are some additional comments for consideration.

Line 158, "incubated at 30 C" for how long?

Line 163, Only soluble lignin was determined? What about insoluble lignin? Also, the equation has two "Total soluble lignin", could the author explain the differences? It would be benefitial to add explanation on "Total soluble lignin", "0.016" below the equation.

Line 207-209, Still confused with this statement about "higher moisture correlates with the release of fermentable sugars". Could the author explain why or give any examples of high moisture biomass lead to higher sugar yield?

Table 3, Substrate concentration should not be "g", please correct. Is this "Lignin" soluble lignin or insoluble lignin, please specify

Line 475, this is not an enzyme, please correct.

Figure 6A, Figure 6 A, why the initial glucose concentration is different? What was the glucose concentration at 0 hrs for each treatment? Please enlarge the legends in the figure as it's difficult to see.

Overall,

There were some design problems with the study. Firstly, the acid pretreatment method should be targeting at reducing the lignin content in the biomass, optimizing the pretreatment conditions to obtain biomass of lowest lignin content (lowest insoluble lignin in the solid and highest soluble lignin in the liquid). The biomass selected should have low-lignin before proceeding to enzymatic hydrolysis. Therotically, the low-lignin biomass would yield more glucose and xylose during enzymatic hydrolysis. Secondly, the content of glucose and xylose shoudn't be used as criteria for successful acid pretreatment. Because of high lignin content in most lignocellulosic biomass, the acid pretreatment function is to break and deconstruct lignin, while the enzymatic hydrolysis only deal with breakdown of cellulose and hemicellulose. Therefore, The authors should consider selecting biomass with the lowest insoluble lignin (or highest soluble lignin) to proceed with enzymatic hydrolysis and fermentation.

Author Response

Response to Reviewer 1

The authors thank the reviewer for his additional comments. Authors respond to each comment (in the red font by mentioning the line numbers) and have made tracked changes to the script.

Comments

The language of the manuscript has been improved. However, the commonts were simply addressed by saying "please see the revised version" without giving a direct location in the manuscript where it has been modified. Below are some additional comments for consideration.

We mentioned line numbers in this round of revision.

Line 158, "incubated at 30 C" for how long?

The incubation time has been added. Please see L165 in the revised script.

Line 163, Only soluble lignin was determined? What about insoluble lignin? Also, the equation has two "Total soluble lignin", could the author explain the differences? It would be benefitial to add explanation on "Total soluble lignin", "0.016" below the equation.

No, both soluble and insoluble fractions of lignin were determined – please see L168-171. The equation has been modified to make to more understandable and equation legends has been mentioned. 0.016 g was the total amount of biomass used to determine the lignin contents – it has been mentioned in the equation legend. Please see Line 170-173.

Line 207-209, Still confused with this statement about "higher moisture correlates with the release of fermentable sugars". Could the author explain why or give any examples of high moisture biomass lead to higher sugar yield?

Actually, we believe that the given statement is correct for microbe-assisted hydrolysis of the biomass but has nothing to do with pretreatment. So, we omitted the sentence. Thank you for bringing this up again. Please see L219-222.

Table 3, Substrate concentration should not be "g", please correct. Is this "Lignin" soluble lignin or insoluble lignin, please specify.

This was the amount of substrate not the concentration. Therefore, we deleted the word “conc” and substituted it with “amount” from Table 3 and wherever was applicable in the script. Please see L394.

Line 475, this is not an enzyme, please correct.

The oversight has been corrected and the world hemicellulose has been changed to hemicellulase. Please see L476.

Figure 6A, Figure 6 A, why the initial glucose concentration is different? What was the glucose concentration at 0 hrs for each treatment? Please enlarge the legends in the figure as it's difficult to see.

In this enzymatic hydrolysis, the biomass from trial 2 was used that had 0.163 g mL-1 of glucose. So, presumably, all the enzymatic hydrolysis had the same amount of glucose at 0 hrs. Additionally, different units of cellulase were added and higher enzyme units yielded more efficient hydrolysis of the biomass, so do the glucose hydrolysis. Aside, to make the things more understandable, some changes were made at line 181-183.

The legend of the figure 6A have been enlarged. Please see the updated figure 6A.

Overall,

There were some design problems with the study. Firstly, the acid pretreatment method should be targeting at reducing the lignin content in the biomass, optimizing the pretreatment conditions to obtain biomass of lowest lignin content (lowest insoluble lignin in the solid and highest soluble lignin in the liquid). The biomass selected should have low-lignin before proceeding to enzymatic hydrolysis. Therotically, the low-lignin biomass would yield more glucose and xylose during enzymatic hydrolysis. Secondly, the content of glucose and xylose shoudn't be used as criteria for successful acid pretreatment. Because of high lignin content in most lignocellulosic biomass, the acid pretreatment function is to break and deconstruct lignin, while the enzymatic hydrolysis only deal with breakdown of cellulose and hemicellulose. Therefore, authors should consider selecting biomass with the lowest insoluble lignin (or highest soluble lignin) to proceed with enzymatic hydrolysis and fermentation.

We completely agree with the reviewer on this point. However, we assume that there is some misunderstanding, which could be due to a lack of proper explanation in the script. We agreed that the amount of insoluble lignin, not the amount of glucose and xylose, is the criterion for enzymatic hydrolysis. Nonetheless, we chose trial 2 for enzymatic hydrolysis for two reasons: first, it had relatively low insoluble lignin, and second it had the higher amount of reducing sugars, assuming that combining both would result in a higher amount of reducing sugars. We tried to mention the similar in the script – please see L389-391.

Nonetheless, we appreciate the reviewer's keen observation of this point, and it would be interesting to compare this point by designing a small project, if this has some effect on ethanol production.

Reviewer 2 Report

the author has made enough changes in the manuscript, No more comments

Author Response

Response to Reviewer 2

The authors thank the reviewer for his more comments. Authors respond to each comment (in the red font by mentioning the page and line numbers) and have made tracked changes to the script.

The kind reviewer asked for the improvement of the Introduction and conclusion section. Some essential information has been entailed in the introduction, as per suggestion. Additionally, the conclusion has been updated. Please see the revised version. Hopefully, this will suffice.

Round 3

Reviewer 1 Report

No more comments.